# Tuning Thermal, Morphological, and Physicochemical Properties of Thermoplastic Polyurethanes (TPUs) by the 1,4-Butanediol (BDO)/Dipropylene Glycol (DPG) Ratio

**DOI:** 10.3390/polym14153164

**Published:** 2022-08-03

**Authors:** Juliano R. Ernzen, Carlos H. Romoaldo, Cedric Gommes, José A. Covas, Angel Marcos-Fernández, Rudinei Fiorio, Otávio Bianchi

**Affiliations:** 1Mantoflex Poliuretanos, Caxias do Sul 95045175, Brazil; julianoernzen@mantoflex.ind.br; 2Chemical Engineering Department, University of Caxias do Sul, Caxias do Sul 95070560, Brazil; chromoaldo@gmail.com; 3Department of Chemical Engineering, University of Liège, B6C, Allée du Six Août 3, B-4000 Liège, Belgium; cedric.gommes@uliege.be; 4Institute for Polymers and Composites (IPC), University of Minho, Campus de Azurém, 4800-058 Guimarães, Portugal; jcovas@dep.uminho.pt; 5Elastomers Group, Institute of Polymer Science and Technology (ICTP-CSIC), Juan de la Cierva, 3, 28006 Madrid, Spain; 6Faculty of Science and Engineering, Maastricht University, 6200 MD Geleen, The Netherlands; r.fiorio@maastrichtuniversity.nl; 7Department of Materials Engineering (DEMAT), Federal University of Rio Grande do Sul (UFRGS), Porto Alegre 90040040, Brazil

**Keywords:** polyurethane thermoplastics, chain extender, reactive extrusion, transparency

## Abstract

Thermoplastic polyurethanes (TPUs) are versatile polymers presenting a broad range of properties as a result of their countless combination of raw materials—in essence, isocyanates, polyols, and chain extenders. This study highlights the effect of two different chain extenders and their combination on the structure–property relationships of TPUs synthesized by reactive extrusion. The TPUs were obtained from 4,4-diphenylmethane diisocyanate (MDI), polyester diols, and the chain extenders 1,4-butanediol (BDO) and dipropylene glycol (DPG). The BDO/DPG ratios studied were 100/0, 75/25, 50/50, 25/75, and 0/100 wt.%. The TPUs were characterized by size exclusion chromatography (SEC), Fourier-transform infrared spectroscopy (FTIR), differential scanning calorimetry (DSC), small-angle X-ray scattering (SAXS), UV–vis spectroscopy, and physical-mechanical properties. The results indicate that DPG promotes compatibility between rigid (HS) and flexible (SS) segments of TPUs. Consequently, increasing DPG content (>75 wt.%) reduced the organization of the rigid segments and the degree of phase separation, increasing the polydispersity of the interdomain distance and the transparency in the UV–visible spectrum of the TPUs. Furthermore, increasing DPG content also reduced the amount of hydrogen bonds present in the rigid phase, reducing or extinguishing its glass transition temperature (T_gHS_) and melting temperature (T_m_), and increasing the glass transition temperature of the flexible phase (T_gSS_). Therefore, increasing DPG content leads to a deterioration in mechanical properties and hydrolysis resistance.

## 1. Introduction

Segmented thermoplastic polyurethanes (TPUs) are copolymers formed by thermodynamically incompatible segments—a soft segment (SS), usually derived from a macrodiol, and a hard segment (HS), formed by a diisocyanate and a low-molecular diol (chain-extender). The microphase separation between HS and SS segments depends on the reactants employed in the TPU synthesis and determines its morphology and physicochemical properties [1,2]. Most TPUs are derived from 4,4-diphenylmethane diisocyanate (MDI) and 1,4-butanediol (BDO) forming the hard segments. Generally, these TPUs are visually opaque due to phase separation and the formation of crystalline domains in the rigid phase [3,4,5].

The transparency in TPUs is an essential characteristic for some products such as coatings and hoses, air filters for capturing particulate matter pollutants, and others [6,7,8,9]. Previous studies have been conducted in order to obtain transparent TPUs. It is known that the chemical structure of the diisocyanate, the type and molecular weight of the soft segment, the degree of crystallinity, and intermolecular interactions (hydrogen bonding) affect the phase separation and transparency of polyurethane materials [9,10]. It is possible to obtain transparent TPUs by decreasing the crystallinity using crosslinkers instead of chain extenders or mixed chain extenders [5,11]. Another possible approach to increase the transparency in TPUs is by modifying the structure of the soft segments (type of polyol) to enhance the miscibility between soft and hard segments [3]. Furthermore, polyether macrodiols are frequently used in transparent PU formulations since these materials show low crystallinity [9]. Higher transparency in TPUs was also achieved by the addition of a non-symmetric isocyanate (isophorone diisocyanate—IPDI), increasing the amorphous phase content [4]. Moreover, decreasing the equivalent weight of the polyol promotes a reduction in the size of the hard-segment length distribution and increases transparency [5]. However, obtaining transparent TPU parts with wall thickness above 2 mm is challenging since these walls tend to become opaque when cooled from the melt due to the inherent low heat transfer in bulk, enabling structural reorganization of the polymer chains. Wang et al. [12] studied self-healing polyurethane films (thickness = 600 μm) based on poly(tetramethylene-ether) glycol, m-xylylene diisocyanate, 1,6-hexanediol, and bis(2-hydroxyethyl)disulfide, and observed optical transparency higher than 90% for all films. Based on the mechanical and optical properties, it was suggested to utilize these polyurethanes in advanced optical devices. An exciting application of transparent TPUs is manufacturing fibers for atmospheric pollution control [6]. Furthermore, TPUs are attractive substrates since they can be spun and combined with emerging particles such as carbon dots showing luminescence to be used as sensors [13].

Xiang et al. prepared a series of TPUs using bisphenol A (BPA) as a chain extender and observed a significant increase in the transparency of the TPUs due to the difficulty in crystallizing the hard segments promoted by BPA [8]. However, BPA must be used with care, because although some studies show that exposure to low doses of BPA has no toxicological effects [14], others maintain that BPA can act as toxic endocrine-disrupting substance [15]. This limits its use in toys and other objects that may be in contact with liquids transport or bodily fluids. Thus, in the present work, we propose an alternative to BPA based on dipropylene glycol (DPG), a lower-toxicity chain extender and a feedstock widely used in the pharmaceutical industry [16].

The main contribution of this work is the elucidation of the effect of DPG as a partial or complete substituent of 1,4-butanediol (BDO) on the morphological properties of TPUs synthesized by reactive extrusion and on molded parts with wall thickness above 1 mm. New TPUs containing different DPG/BDO ratios were synthesized by reactive extrusion, and their morphology and thermal and physicochemical properties were evaluated. In addition, a deep investigation of the morphological characteristics based on the microphase separation between soft and hard segments was conducted.

## 2. Materials and Methods

The isocyanate used to synthesize the TPUs was 4,4′-diphenylmethane diisocyanate (MDI), purchased from Dow Chemical, São Paulo, Brazil (ISONATE 125M, NCO content = 33.5 wt.%, ⍴ = 1.23 g.cm^−3^). Two linear polyols polyesters (Elapol 5010A, molecular weight of 1000 g.mol^−1^, hydroxyl index of 110–114 mgKOH.g^−1^, ⍴ = 1.15 g.cm^−3^; and Elapol 3020A, molecular weight of 2000 g.mol^−1^, hydroxyl index of 54–58 mgKOH.g^−1^, ⍴ = 1.15 g.cm^−3^) were purchased from Elachem, Vigevano, Italy. The chain extenders used were 1,4 butanediol (BDO, ⍴ = 1.02 g cm^−3^) and dipropylene glycol (DPG, a mixture of isomers, ⍴ = 1.02 g.cm^−3^), purchased from Dow Chemical, São Paulo, Brazil.

### 2.1. Reactive Extrusion

A 75/25 wt.% blend of the two polyols (Elapol 3020A/Elapol 5010A; 2000/1000 g.mol^−1^, respectively) was prepared beforehand at 80 °C. This blend was kept under stirring for 15 min before starting the reactive extrusion. A pre-polymer was prepared in a 5 L stainless steel vessel by adding MDI and incorporating the polyol blend. This mixture was kept at 80 °C and stirred during the reactive extrusion step. The systems were maintained in an inert atmosphere (N_2_) and coupled to a heated metering pump at the same temperature. All TPUs were synthesized with ~50 wt.% HS with a 1.05/1 NCO/OH functional groups ratio. TPUs containing different BDO/DPG ratios (100/0, 75/25, 50/50, 25/75, and 0/100 wt.%) were prepared. The pre-polymer and chain extenders (BDO, DPG, and their mixtures) were dosed in order to obtain a constant production rate of approximately 2 kg.h^−1^ of TPU in an LTE 16–48 co-rotating twin-screw extruder (LabTech Engineering Company Ltd., Thailand) with a length-to-diameter ratio (L/D) of 48. Processing temperatures ranged from 160 to 220 °C, and the screw speed used was 250 rpm. The screw profile was optimized for TPU production based on the studies of Amin et al. [17], presenting two mixing sections with kneading elements staggered at 30, 60, and 90°, separated by conveying elements. These processing conditions resulted in an average residence time of approximately 300 s.

The extruded polymers were cooled in water (25 °C) and pelletized for subsequent characterization. In addition, the materials were injection molded (at 250 °C and 90 bar) to obtain specimens for the SAXS, UV–vis, and physical-mechanical tests.

### 2.2. Size Exclusion Chromatography (SEC)

The molecular weight of the TPUs was determined by size exclusion chromatography (SEC) in a Perkin-Elmer series 200 (refraction index detector) chromatograph using dimethylformamide with 1% of BrLi as eluent. Other parameters used for this characterization were as follows: sample concentration of 10 mg.mL^−1^, flow rate of 1 mL.min^−1^, injected volume of 10 μL, and column temperature of 35 °C. Polystyrene standards were used for the calibration curve.

### 2.3. Fourier Transform Infrared Spectroscopy (FTIR)

The chemical structure of the TPUs and the effect of the chain extender variation on phase separation were evaluated by Fourier transform infrared spectroscopy (FTIR, Perkin-Elmer Spectrum 400 spectrometer, Waltham, EUA) in attenuated total reflection mode (ATR; diamond crystal at 45°). The samples were scanned 32 times in the range 4000–450 cm^−1^ at a resolution of 2 cm^−1^. In addition, the degree of hydrogen bonding and the amount of hard segments dispersed in the soft phase were computed by mathematical deconvolution [18].

The region between 1500 and 1800 cm^−1^ of the FTIR absorption bands (carbonyl C=O group) is sensitive to the amount and type of hydrogen bonding in polyurethane materials. Aiming to clarify the relative quantities of the different types of hydrogen bonding present in the samples, we opted to evaluate the interactions in the polyester polyol (PM) used. The absorption band at ~1745 cm^−1^ (band 1 PM) is assigned to the corresponding free carbonyl groups; the absorption band at ~1730 cm^−1^ (band 2 PM) is associated to the carbonyl groups physically bonded by dipole-dipole interactions; and the adsorption band at ~1700 cm^−1^ (band 3 PM) is related to the carbonyl groups physically attached by hydrogen bond interactions (between the terminal -OH groups of the macrodiol and the C=O of ester groups) [19]. The results of the deconvolution of each absorption band are shown in the Appendix A. The definition of the contributions of the bands of the flexible segment provides a better understanding of the phase separation in the TPUs.

Additionally, we performed immersion of the polyol samples in water and ammonia. In water, the absorption area of the band 3 PM increased compared to the other bands. This change shows that the water molecules created new hydrogen bonds with the carbonyl groups of the polyester-diol, changing to some extent the dipole–dipole interactions. However, to verify the nature of the hydrogen-bonded carbonyl group of PM occurring in polyurethane, which is necessary for the proper identification of the position of each peak found in the TPUs, N-H proton donor group from the ammonia solution was introduced into PM macromolecules. The ammonia solution was expected to provide stronger hydrogen interactions than water, moving the band 3 PM to lower wavenumbers and reproducing the condition when the urethane group NH (located in the hard segments) would interact with the group C=O (located on the soft segments), through hydrogen bonds in TPUs. The formation of a new band at ~1650 cm^−1^ (band 4 PM), corresponding to the interaction of the N-H groups from the ammonia solution with the C=O of the macrodiol, was observed. The amount of hydrogen bonds and its distribution in TPU phases modify the phase separation nature of PUs [20].

To evaluate the effect of the chain extenders and determine the rigid and flexible phase fractions, contributions of six bands in the region 1500–1800 cm^−1^ were considered [18,19,21]. The evaluation of these bands has been conducted in previous studies to determine the relative amount of hydrogen bonds in petrochemical and biobased polyols (e.g., carbonate-based polyol [4,18,19]). In this study, mathematical adjustments in the carbonyl region were performed for each spectrum considering six bands assuming Gaussian functions (see Appendix A). The six bands are described as follows: (I) at 1675 cm^−1^, H-bonded urethane carbonyl groups (ordered phase—hard segments); (II) at 1699 cm^−1^, H-bonded urethane carbonyl groups (disordered phase—hard segments); (III) at 1710 cm^−1^, H-bonded carbonyl groups (soft-hard segment); (IV) at 1722 cm^−1^, free carbonyl from urethane groups (hard segment); (V) at 1732 cm^−1^, carbonyl–carbonyl interactions (soft segments); and (VI) at 1745 cm^−1^, free carbonyl (soft segment). The chain extender effect in the TPUs was also quantitatively evaluated by FTIR, determining the weight fraction of hydrogen-bonded urethane groups (*X_b_*), the weight fraction of hard segment dispersed in the soft segment (*W_h_*), mixed-phase weight fraction (*MP*), soft phase weight fraction (*SP*), and hard phase weight fraction (*HP*). Equations (1)–(5) were used for the determination of *X_b_*, *W_h_*, *MP*, *SP*, and *HP*, respectively [19]:(1)$$X_{b}=\frac{A_{b}}{k^{\prime }\;A_{f}+A_{b}}=\frac{A_{1675}+A_{1699}}{k^{\prime }A_{1722}+A_{1675}+A_{1699}}$$
(2)$$W_{h}=\frac{\left(1-X_{b}\right)f}{\left[\left(1-X_{b}\right)f+\left(1-f\right)\right]}$$
(3)$$MP=fW_{h}$$
(4)SP=MP+(1−f)
(5)HP=1−SP
where $A_{b}$ is the area associated with H-bonding between two urethane groups (bands I and II), $A_{f}$ is the absorbance of free carbonyl from non H-bonded urethane groups (band IV), $k^{\prime }$ is a constant equal to 1.2 (extinction coefficient) [18], and f is the weight fraction of hard segments in the polymer, determined from the initial molar ratios.

### 2.4. Differential Scanning Calorimetry (DSC)

The thermal transitions of the polyols and TPUs were determined by differential scanning calorimetry (DSC, Netzsch DSC 204 Phoenix, Selb, Germany) under nitrogen atmosphere (50 mL.min^−1^). Samples of 9–10 mg were used. The melting temperature and enthalpy were calibrated with indium, tin, bismuth, and zinc standards. The samples were analyzed from −80 to 240 °C at a heating rate of 20 °C.min^−1^. The value of the T_g_ is given at the onset point, and the variation of heat capacity ΔC_p_ is calculated at the ΔT_g_ midpoint. To determine the specific transition temperatures of the rigid phase, model reactions were carried out with the MDI-BDO and MDI-DPG at 80 °C under mechanical stirring for 1 h.

### 2.5. Small-Angle X-ray Scattering (SAXS)

Small-angle X-ray scattering (SAXS) experiments were performed on the SAXS1 beamline of the Brazilian Synchrotron Light Laboratory (LNLS), monitored with a photomultiplier, and detected by a Pilatus detector (300 k, 84 mm × 107 mm) positioned at a distance of 836 mm. The generated scattering wave vectors (q) ranged from 0.13 to 2.5 nm^−1^. The wavelength of the incident X-ray beam (λ) was 0.155 nm. Sample dimensions were 3 mm in diameter and thickness of 1 mm. A silver behenate (AgBeH) standard was used to calibrate the diffraction angle. All measurements were performed at 23 °C. The background and parasitic scattering were determined in separated measurements using an empty holder and were subtracted from the samples scattering.

### 2.6. Light Transmittance

The light transmittance of the TPU materials was evaluated using a Thermo Scientific Evolution 60 UV–vis spectroscope (Waltham, EUA) with a wavelength range of 200–900 nm using injection-molded bars (3 mm thickness).

### 2.7. Physical-Mechanical Properties

The physical-mechanical properties of the samples were evaluated in terms of density, abrasion resistance, compression set, hardness, and tensile properties. Tensile tests were performed according to ISO 527-2/1BA/500 using an Instron (EMIC DL 30,000 with extensometer) with a crosshead speed of 500 mm.min^−1^. The tensile test was also performed before and after hydrolytic degradation. The hydrolysis experiments were done according to ASTM D3137-81. The samples were suspended in a container with 5 L of deionized water for 96 ± 1 h at a temperature of 85 ± 1 °C. The tensile properties were compared before and after hydrolysis.

Density measurements were performed according to ASTM D792-13. The abrasion resistance was analyzed according to ASTM D5963-04. The mass loss was calculated after 40 m with a P60 abrasive paper. TPU compression set at 23 °C and 70 °C was carried out according to ASTM D395-16. Shore A hardness was determined in a durometer Woltest SD 100 according to ASTM D2240-15.

## 3. Results and Discussion

The size exclusion chromatography results are presented in Table 1. Molecular weights achieved were high. The modification of the ratio BDO/DPG did not promote a significant change in either number average molecular weight (M_n_) and weight average molecular weight (M_w_), nor in the molecular weight distribution (M_w_/M_n_). As the rigid phase content and the NCO/OH ratio were kept constant for all samples, assuming the same reactivity for both chains extenders, the similarity among the SEC results was expected. Comparable values for M_n_, M_w_, and M_w_/M_n_ in TPUs were found in previous studies [22].

Generally, when MDI is used as a diisocyanate, a less pronounced degree of microphase separation is observed compared to symmetric aliphatic diisocyanates [21]. This fact is associated with the MDI’s structural rigidity and low molecular mobility, resulting from the presence of aromatic rings in the structure, which promote additional interactions between the nitrogen of the urethane group non-associated by hydrogen bond with the electrons of the MDI rings [23].In order to evaluate the solubility of the TPU phases, 2D plots of solubility parameters (Bagley plots) were made. The Bagley methodology has been successfully applied to evaluate the miscibility of polymer blends [24,25]. This methodology is based on the assumption that the effects of the dispersion (*δ_d_*) and polar (*δ_p_*) components of the solubility parameter show close similarity, while the effect of the hydrogen-bonding (*δ_h_*) component is of an entirely different nature. Dispersive, polar, and hydrogen contributions were computed using the Hoftyzer–Van Krevelen group contribution method [26]. Accordingly, Bagley introduced the parameter *δ_v_*, given by:(6)$$\delta_{v}^{2}=\delta_{d}^{2}+\delta_{p}^{2}$$

The Bagley methodology (Equation (6)) assumes that only two solubility parameters are needed to describe the solubility parameter (*δ*) for a solvent: one corresponding to the “physical” interactions (polar and non-polar effects, *δ_v_*), and the other to the “chemical” interactions (hydrogen bonding effects, *δ_h_*). The solubility region can be determined by plotting *δ_h_* vs. *δ_v_*. The distance (*D*_12_) between a pair in a Bagley plot is given by [25]:(7)$$D_{12}=\sqrt{{\left(\delta_{1,v}-\delta_{2,v}\right)}^{2}+{\left(\delta_{1,h}-\delta_{2,h}\right)}^{2}}$$

The miscibility region in a Bagley plot for systems with van der Waals forces is delimitated by a semicircle with an “interacting radius” (*D*_12_) of around 2.5 MPa^1/2^. This region can be expanded to 5.5 and 7 MPa^1/2^ for other systems presenting hydrogen bonding [24].

Figure 1a shows the Bagley plot, with the polyester polyol (2000 g/mol) in the circle center. In this figure, we evaluate the possible solubility of the components of the flexible phase. The two polyols present miscibility since the distance *D*_12_ is small (0.4). However, the solubility of the chain extenders in the polyols is low. For BDO, a *D*_12_ value of 11.6 is calculated, while for DPG, this value is 8.5. In Figure 1b, the circle center is MDI, and this figure compares the distances between the MDI and the two chain extenders, that is, the possible compositions of rigid phases. As for the polyols, the distance *D*_12_ concerning the DPG is also smaller (10.1) than that found for BDO (13.0). This result reflects the lower polarity of this chain extender, which might induce the formation of mixed phases between the rigid and flexible segments. Higher compatibility will increase mixing entropy and smoother phase separation between the blocks. Gallu and coworkers [27] used Hansen’s solubility parameters to elucidate the phase separation in TPU formed by a rigid segment of MDI-BDO and a flexible segment formed by a fatty acid. They confirmed that the segments show incompatibility in situations where the solubility parameters have differences *δ* higher than 3.6 MPa^1/2^ and a *D*_12_ of ~7.1, which were evidenced by DSC and SAXS measurements.

After reactive extrusion, the hydroxyl bands characteristic of polyol (OH) at 3600–3300 cm^−1^ and isocyanate (NCO) at 2277 cm^−1^ were not found, which indicates that these groups observed (Appendix A) at 1317, 1500, and 1520 cm^−1^ are attributed to stretching modes of CN, CH_2_, CH_3_, and NH_2_. The band at 3324 cm^−1^ is assigned to the free N–H stretching vibration of the urethane groups, while the urea group’s formation is evidenced through the stretching mode (NH_2_) at 1654 cm^−1^ and the benzene ring domain in MDI (Appendix A) (band at 1600 cm^−1^) [19].

From the FTIR spectra (carbonyl region 1600–1800 cm^−1^), the weight fraction of hydrogen-bonded urethane groups (*X_b_*), weight fraction of hard segments dispersed in the soft segments (*W_h_*), mixed phase weight fraction (MP), soft phase weight fraction (SP), and hard phase weight fraction (HP) were determined, and they are presented in Table 2. For all TPUs, a high fraction of hydrogen-bonded urethane groups (parameter *X_b_*—Equation (1)) was noted regardless of the type of chain extender used in the synthesis. It was also observed that the increase in DPG reduced this parameter to some extent because DPG is a mixture of isomers, which hinders the formation of hydrogen bondings. Therefore, the *W_h_* fraction is higher in TPUs with higher DPG content.

Increasing DPG content resulted in a slight increase in MP. According to theoretical calculations, several mixed phases occur for polyester or polyether polyurethane [28]. The results shown in Table 2 agree with previous results reported in the literature for TPUs showing highly separated phases [29].

The differential scanning calorimetry (DSC) curves of the polyol blend, representing reaction products of MDI-BDO, MDI-DPG, and MDI-BDO/DPG (hard segment models), are presented in the Appendix A. The polyol blend showed a T_g_ at −18.9 °C and a T_m_ at 40 °C. For all hard segment models, a T_g_ was noted between 102 and 112 °C. The MDI-BDO sample showed a T_m_ at 214 °C, which agrees with the literature [30]. Note that for MDI-DPG, no melting temperature was observed, while for the BDO/DPG 50/50 blend, a reduction in the quantity of crystallizable material was noticed. Thus, it is evident that DPG suppresses the crystallization of the rigid phase, which is an effect similar to that observed with TPUs with BPA [8].

The DSC results of the TPUs are shown in Figure 2. Three main events were observed: the glass transition temperature of the flexible phase (T_gSS_), the glass transition temperature of the rigid phase (T_gHS_), and the melting temperature (T_m_). T_gSS_ was the only transition observed for all samples, and the increase in the amount of DPG promoted a gradual increase in the temperature of this transition from 0.5 to 24.1 °C (BDO/DPG 100/0 to 0/100, respectively). This increase is related to increased miscibility between rigid and flexible segments [3,31]. Hesketh et al. [32] observed higher T_gSS_ values with the increased amount of HS dispersed in the flexible phase. Finnigan et al. [33] associated a decrease in T_gSS_ with an increase in the molecular weight of the flexible segment, which led to an increase in phase separation. The DPG increases the entropy of TPU, and the higher miscibility between the rigid and flexible phases allows the formation of hydrogen bonds with the flexible phase (ester groups), which promotes a decrease in the mobility of this phase with a consequent increase in T_gSS_. Furthermore, the polyols used in the present study showed glass transition temperatures (T_g_) of ca. −19 °C, closer to the T_gSS_ of the samples BDO/DPG 100/0 and 75/25, indicating that these two TPUs present higher degrees of phase separation. For the other samples, an increase in DPG promoted an increase in T_gSS_, indicating a decrease in phase separation since an increase in the miscibility between the rigid and the flexible segments increases the difference between the T_gSS_ of the TPUs and the T_g_ of the pure polyols [32,34].

In samples BDO/DPG 100/0 and BDO/DPG 75/25, the glass transition temperature of the rigid phase (T_gHS_) and the melting (T_m_) of the crystallites was evidenced (Figure 2). These values are in agreement with the literature [3,32]. The sample TPU 100/0 BDO/DPG showed a T_gHS_ of 106.5 °C, while the TPU BDO/DPG 75/25 presented a slightly lower T_gHS_ of 102.3 °C. This reduction in T_gHS_ can be related to increased miscibility between the hard and soft domains. The Tm results also confirmed the miscibility between the phases. Still, the sample BDO/DPG 100/0 presented a T_m_ of 176 °C (melt enthalpy, ΔH_m_ = 17.2 J.g^−1^), and the addition of DPG (sample 75/25 BDO/DPG) induced a slight decrease of T_m_ to 171 °C (ΔH_m_ = 16.4 J.g^−1^). Subsequent increments in the DPG content (samples BDO/DPG 50/50, 25/75, and 0/100) prevented the formation of crystalline domains, indicating an increase in the entropy of the rigid phase and confirming the higher miscibility between phases. All these results corroborate those obtained by FTIR.

Figure 3 shows that ΔC_p_ at the T_g_ of the SS decreases with the DPG content in the TPU. This decrease can be explained by the change in the soft segment involved in the transition. Based on the assumptions of Camberlin and Pascault [35], the degree of phase separation was estimated [27]. The TPU with BDO, as expected, presents greater phase separation (~70%), and increasing the DPG content leads to a linear reduction of phase separation, which suggests an increase in affinity between the SS segments and HS. This effect was evidenced by the Bagley diagram (Figure 1) and the increase in T_gSS_ values above 50% of DPG.

The small-angle X-ray scattering results are presented in Figure 4. All samples presented a scattering peak, except the sample BDO/DPG 0/100 (Figure 4a). The peaks observed indicate microphase separation related to the supramolecular structures found in the rigid segments of TPU [36,37]. Sample BDO/DPG 0/100 does not present any peak since DPG isomers reduce the order of the rigid segments and, consequently, the degree of phase separation [36]. The structural parameters were based on the extrapolation of the SAXS curves using Porod’s law. Therefore, it was considered that the scattering objects are periodic stacks consisting of alternate structures [30,34,38,39].

In order to estimate the interdomains distance, the Lorentz correction was applied to the SAXS results [30,34,38,39]. In addition, the polydispersity of the interdomains distance, representing the distribution of the interdomains distance from an average value, was also investigated from the 1D correlation function (Figure 4b) [30,38]. This calculation assumes a lamellar morphology, which is not likely because at a hard segment content of approximately 45% weight in these TPUs, a gyroid morphology is more likely. However, with this calculation, comparison between TPUs can be done. The average interdomains distance and polydispersity of the rigid segment’s results are summarized in Table 3.

According to the results presented in Table 3, adding DPG up to the fraction 50/50 BDO/DPG did not cause a drastic modification of the interdomains distance nor of polydispersity. However, the sample BDO/DPG 25/75 showed a very significant increase in polydispersity. In addition, this sample did not show a modification in the average interdomains distance. Hence, the sample BDO/DPG 25/75 shows a considerable variation in the interdomains distance of the rigid segments with a lower degree of phase separation. Furthermore, it was impossible to determine the average interdomains distance or the polydispersity for the sample BDO/DPG 0/100 since it did not present a peak in the SAXS analysis. Therefore, SAXS results corroborate those found from the DSC and FTIR analyses.

Evaluating the data regarding the electronic density variation between the phases ${\left(\rho_{HS}-\rho_{SS}\right)}^{2}$ and the invariant (Q), one can observe a reduction in the contrast between the phases and, consequently, in the degree of phase separation (see Appendix A). The estimated electronic density value for the SS phase is 388.8 e^−^.nm^−3^, and the electronic density value for the rigid phase can be expressed as a linear function of the DPG content ($\rho_{HS}$ = 0.82 × DPG (wt.%) + 327.6, R = 0.986). For the sample BDO/DPG 25/75, a smaller variance in the electronic density is observed compared to the other samples, suggesting a higher degree of mixing between the phases. This increase in mobility may promote an increment in the fraction of organized domains of the rigid segment, as observed from the thickness of the hard microdomain (T_HS_; see Table 3).

Hence, the dissolution of the hard segments in the soft matrix can lead to a mixture of micro-phases, affecting the micro-domain thickness. The stacked lamellae of the longer sequences of rigid segments, surrounded by the shorter sequences of rigid segments (BDO) dissolved in the soft matrix, may express the limit of the diffuse phase. Considering that the BDO and the DPG present similar sizes, there is a balance between solubility and steric hindrance that substantially contributes to phase separation since the dimension of the domains is strongly affected by their ability to dissolve in the soft matrix [20,40,41]. As a result, the polymer’s crystallinity and mechanical properties are modified.

The phase morphology of the polyurethanes was reconstructed by the clipped Gaussian random field (GRF) model [42], and the results are shown in Figure 5. For the TPUs, co-continuous domains were obtained, as expected, due to the amount of hard segment in the TPUs (approximately 45%). Although up to BDO/DPG 25/75, the structure pattern is practically maintained, structural order was noticed for the sample BDO/DPG 0/100. However, the effect of DPG is more pronounced in the polydispersion (see the hard and soft domains distributions in the Appendix A) of the rigid domains through the solvation effect, leading to an increase in the entropy of the mixture. However, this will reflect on the polymer’s properties, such as the mechanical and transparency properties.

Figure 6 shows the light transmittance (wavelengths in the UV–visible region) for the TPUs (3.2 mm) and a glass (used as reference). An increase in DPG content increased the transparency of the TPU. This is related to an increase in the amorphous phase and a decrease in the degree of phase separation, with an expected formation of smaller rigid domains or even their elimination. Previous studies reported an increase in the optical transmittance with a decrease in the crystallinity degree of TPUs based on IPDI and hexamethylene diisocyanate (HDI) [4,43]. Here, some points need to be taken into account. First, the increase in miscibility between the segments reduces the phase separation, and with this, there is an increase in transparency [7]. As a rule, reducing the size of hard domains due to greater compatibility of the phases leads to increased transparency.

The physical-mechanical properties of TPUs are strongly affected by the relative amount of hydrogen bond and crystalline phase [30]. Table 4 summarizes the tensile strength, compression set, density, and abrasion results. Regarding the stress at 100% of deformation, a significant reduction occurs with the increase in DPG content (results shown in the Appendix A). This reduction is accelerated when the amount of DPG is higher than 50% of the ratio BDO/DPG. In the hydrolyzed samples, it is clear that the larger polydispersity of the domains and the reduction of the relative amount of hydrogen bonding have a fundamental role in the hydrolysis resistance. First, hydrolysis occurs in the flexible segments in polyesters, and later, in the rigid segments [44]. However, the urethane groups are more susceptible to chain scission due to the greater amount of mixed phase. Thus, water has more access to urethane bonds due to the smaller number of organized domains. Unfortunately, it was impossible for the sample containing only DPG to be evaluated by the tensile test since the specimen had deteriorated. Given this, it is clear that increasing the amount of mixed phase and dispersion and reducing the hydrogen bond density drastically affect the mechanical properties.

As expected, forming a smaller amount of organized domains makes the sample more susceptible to plastic deformation since the organized phase, being more compact, shows higher resistance to deformation. This behavior is also evidenced in the hardness results. Concerning the abrasion results, there is an initial reduction and later an increase in the abrasion loss with the increase in DPG in the BDO/DPG ratio. Sample BDO/DPG 50/50 shows the highest abrasion resistance, likely because it presents a good balance between stiffness and flexibility, hindering the removal of polymer particles during the abrasion test. It was previously observed that polymers presenting high toughness have high abrasion resistance [45].

## 4. Conclusions

A series of high-molecular-weight thermoplastic polyurethanes based on polyester polyol, MDI diisocyanate, and BDO/DPG chain extenders were successfully synthesized by reactive extrusion.

FTIR analysis of the carbonyl region allowed for determination of the fraction of hydrogen-bonded urethane groups and the percentage of hard segments mixed in the soft segment phase. The increase on DPG decreased the amount of hydrogen-bonded urethane groups and increased the mixing of segments.

Therefore, phase separation, as studied by DSC, decreased with the increase of DPG, resulting in an increase in soft segment T_g_ and a decrease in hard segment crystallinity until disappearance. SAXS analysis proved that polydispersity of the domains increased substantially at DPG ratio content above 50%, with disappearance of the peak due to phase separation at 100% ratio of DPG.

The increase in the compatibility between the rigid and flexible phases and the decrease in hard segment crystallinity of TPUs with the increase in DPG content seems to be the preponderant factor influencing the physical properties. Thus, transparency is increased; tensile strength, hardness, and hydrolytic resistance are decreased; and the compression set is increased.

## Figures and Tables

**Figure 1 polymers-14-03164-f001:**
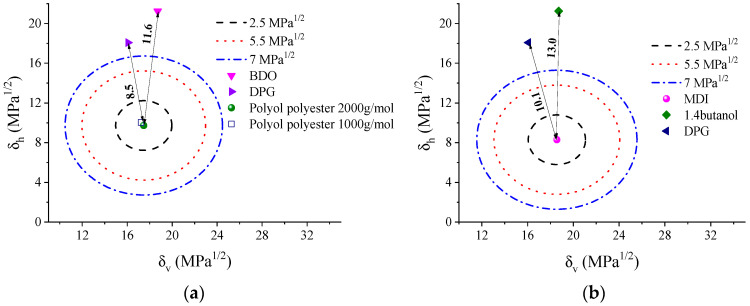
(**a**) Bagley plots for TPU soft phase calculated using constant contribution approximation: (**a**) SS segment (in circle’s center) and (**b**) HS segment (in circle’s center). The dotted circles indicate *D*_12_ equal to 2.5 MPa^½^ (black), 5.5 MPa^½^ (red), and 7 MPa^½^ (blue).

**Figure 2 polymers-14-03164-f002:**
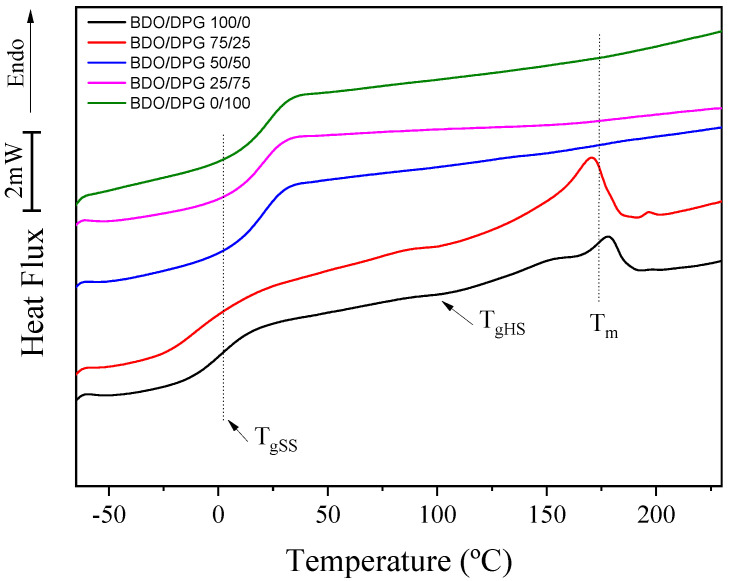
Differential scanning calorimetry results for the TPUs (second heating cycle).

**Figure 3 polymers-14-03164-f003:**
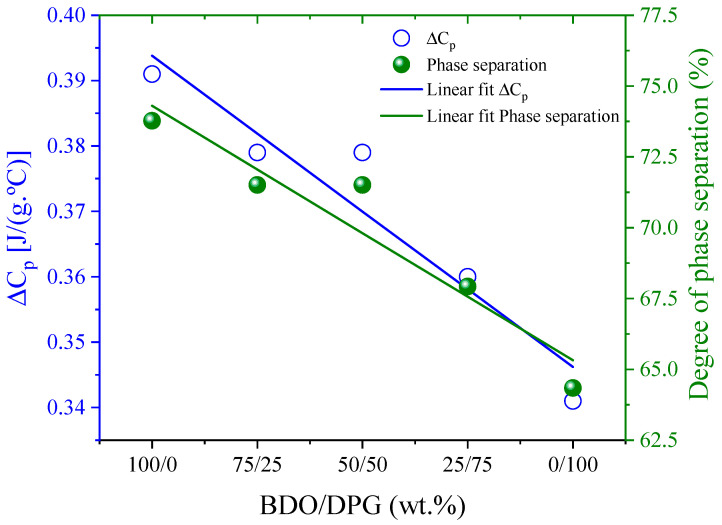
Change of heat capacity, ΔC_p_, at the glass transition of the SS and of the corresponding phase separation degree, as a function of DPG content.

**Figure 4 polymers-14-03164-f004:**
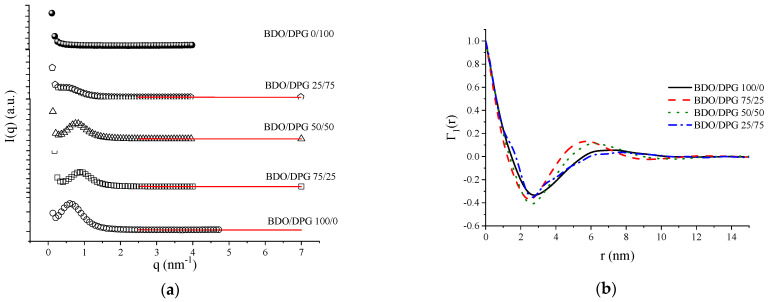
Small-angle X-ray scattering results for the TPUs: (**a**) SAXS profiles and Porod extrapolation (red line); (**b**) 1−D correlation functions (Γ_1_ (r)).

**Figure 5 polymers-14-03164-f005:**
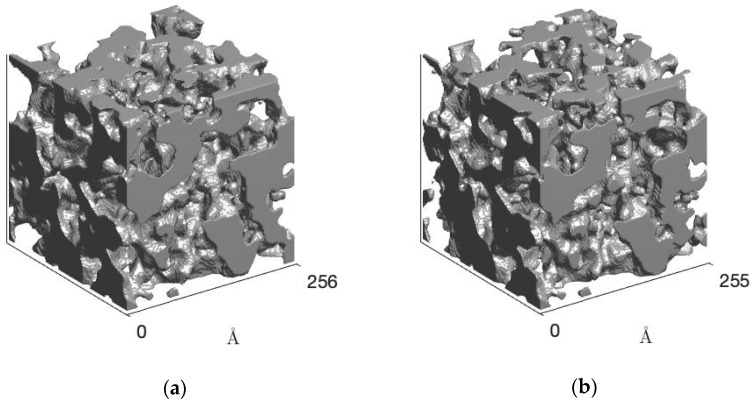
Clipped GRF model obtained from the fitted SAXS curves for TPU BDO/DPG 100/0 (**a**) and BDO/DPG 25/75 (**b**).

**Figure 6 polymers-14-03164-f006:**
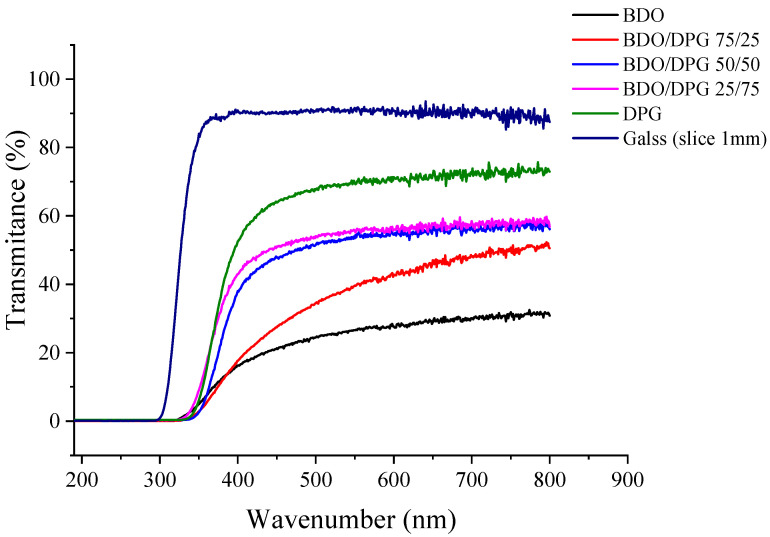
Light transmittance in the UV–visible region for the TPUs and a glass slice.

**Table 1 polymers-14-03164-t001:** Molecular weight (M_n_, M_w_) and molecular weight distribution (M_w_/M_n_) obtained by SEC.

BDO/DPG	M_n_ × 10^−3^ (kDa)	M_w_ × 10^−3^ (kDa)	M_w_/M_n_
100/0	143.2	256.7	1.79
75/25	153.5	255.8	1.67
50/50	160.0	303.3	1.89
25/75	164.1	288.2	1.76
0/100	154.9	229.2	1.48

**Table 2 polymers-14-03164-t002:** Calculated parameters for determination of microphase separation: hard segment weight fraction in the polymer from initial molar ratios (f), volumetric fraction of HS (φ), weight fraction of hydrogen-bonded urethane groups (*X_b_*), weight fraction of hard-segment-dispersed soft segment (*W_h_*), mixed phase weight fraction (*MP*), soft phase weight fraction (*SP*), and hard phase weight fraction (*HP*).

BDO/DPG	*f*	φ	*X_b_*	*W_h_*	MP	SP	HP
100/0	0.469	0.468	0.77	0.17	0.08	0.61	0.39
75/25	0.461	0.439	0.76	0.17	0.08	0.62	0.38
50/50	0.450	0.450	0.74	0.18	0.09	0.62	0.38
25/75	0.446	0.443	0.73	0.19	0.09	0.62	0.38
0/100	0.437	0.439	0.66	0.23	0.11	0.64	0.36

**Table 3 polymers-14-03164-t003:** SAXS parameters obtained by the Lorentz correction and 1D correlation function.

BDO/DPG	L (nm)	L1D (nm)	T_HS_ (nm)	BT (nm)	P	AIT (nm)	AC (nm)	LC
100/0	7.1	6.42	3.00	1.96	2.6	0.59	1.29	0.31
75/25	6.0	5.83	2.56	1.82	2.3	0.56	1.20	0.31
50/50	6.7	6.15	2.77	2.04	2.8	0.64	1.37	0.33
25/75	6.5	7.70	3.41	1.41	10.7	0.06	0.68	0.18
0/100	-	-	-	-	-	-	-	-

L = long period obtained by Lorentz correction; L_1D_ = long period computed by 1D correlation function [38,39]; T_HS_ = thickness of the hard microdomains, assuming a lamellar morphology; BT = average hard block thickness; P = polydispersity; AIT = average interface thickness; AC = average core thickness; LC = local crystallinity.

**Table 4 polymers-14-03164-t004:** Physical-mechanical properties of the TPUs.

BDO/DPG	σ^b^TS (MPa)	σ^a^TS (MPa)	CS_23°C_ (%)	CS_70°C_ (%)	AL (%)	ρ (g.cm^−3^)	Hardness (Shore A)
100/0	31.3 ± 2.6	26.1 ± 1.7	44.8 ± 0.4	79.3 ± 0.1	5.4 ± 0.3	1.24 ± 0.01	95 ± 1
75/25	28.0 ± 1.5	23.3 ± 1.5	45.2 ± 1.1	86.1 ± 1.9	6.8 ± 0.1	1.23 ± 0.01	95 ± 1
50/50	25.1 ± 0.8	21.7 ± 0.9	40.7 ± 2.4	83.2 ± 0.6	3.5 ± 0.1	1.23 ± 0.01	94 ± 1
25/75	22.4 ± 1.0	12.7± 1.3	49.6 ± 1.9	89.6± 0.3	5.8 ± 0.7	1.22 ± 0.01	90 ± 1
0/100	19.3 ± 2.0	-	76.3 ± 1.4	99.1 ± 1.0	8.1 ± 0.6	1.21 ± 0.01	88 ± 1

σ^b^TS = tensile strength before hydrolysis; σ^a^TS = tensile strength after hydrolysis; CS_23°C_ = compression set at 23 °C; CS_70°C_ = compression set at 70 °C; AL = abrasion loss; ρ = density.

## Data Availability

The raw data needed to reproduce these findings can be shared if requested from the authors.

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
