# Peer review of "Tuning Thermal, Morphological, and Physicochemical Properties of Thermoplastic Polyurethanes (TPUs) by the 1,4-Butanediol (BDO)/Dipropylene Glycol (DPG) Ratio"

_polymers, 2022, doi:10.3390/polym14153164_

Round 1
Reviewer 1 Report
This study highlights the effect of two different chain extenders and their combination on the structure-property relationships of TPUs synthesized by reactive extrusion. The whole work introduces the influence of the rigid (HS) and flexible (SS) of TPUs on its performance in detail, and explains the mechanism of this effect in detail by using DSC, SEC, SAXS and other methods. Overall, this work has great significance for TPU performance regulation. In addition, the data enrichment of the whole work can well support the results. I think it can be publish for polymers after minor revise.
(1) The resolution of the pictures in the manuscript needs to be adjusted. (Fig. 1,2,3,4,5,6).
(2) The “Effects of” should be deleting in the title.
(3) There are still some grammatical errors in the manuscript that need to be revised.
(4) Some date results should be introduced in detail in the abstract part.
Author Response
Dear Reviewer,
On behalf of the authors, we thank you for your contributions.
“This study highlights the effect of two different chain extenders and their combination on the structure-property relationships of TPUs synthesized by reactive extrusion. The whole work introduces the influence of the rigid (HS) and flexible (SS) of TPUs on its performance in detail, and explains the mechanism of this effect in detail by using DSC, SEC, SAXS and other methods. Overall, this work has great significance for TPU performance regulation. In addition, the data enrichment of the whole work can well support the results. I think it can be publish for polymers after minor revise.”
(1) The resolution of the pictures in the manuscript needs to be adjusted. (Fig. 1,2,3,4,5,6).
Answer to Reviewer: Thank you for your contribution. All figures have been improved.
(2) The “Effects of” should be deleting in the title.
Answer to Reviewer: Thank you for your contribution. A new title has been adopted
(3) There are still some grammatical errors in the manuscript that need to be revised.
Answer to Reviewer: Thank you for your contribution. The text has been corrected
(4) Some date results should be introduced in detail in the abstract part.
Reply to Reviewer: Thank you for your contribution. The text has been improved
Reviewer 2 Report
The manuscript entitled “Effects of the 1, 4-butanediol (BDO)/dipropylene glycol (DPG) ratio on the morphological, thermal, and physicochemical properties of thermoplastic polyurethanes (TPUs) “ by J. R. Ernzen et al . reports new results concerning the role of dipropyleneglycol as substituent of 1,4-butanediol on the physical properties of thermoplastic polyurethanes .In this order, authors reports analysis by Fourier-transform infrared spectroscopy (FTIR), UV-VIS spectroscopy, small-angle X-ray scattering (SAXS), differential scanning calorimetry (DSC) and mechanical properties. The discussions are well structured. In the revised manuscript, the following comments are necessary to be included:
i) what is the physical rationale for using 3 components in the deconvolution of Figures S1a-d and 4 components in the case of Figure S1e?
ii) Similar question for Figure S2
iii) Authors should to show the FTIR spectra in the 400-4000 cm-1 spectral range and then to focus the discussion on the spectral range 1600-1800 cm-1. A comment concerning other changes observed in the spectral range 400-4000 cm-1 will be welcome.
Author Response
Dear Reviewer,
On behalf of the authors, we thank you for your contributions.
“The manuscript entitled “Effects of the 1, 4-butanediol (BDO)/dipropylene glycol (DPG) ratio on the morphological, thermal, and physicochemical properties of thermoplastic polyurethanes (TPUs) “ by J. R. Ernzen et al . reports new results concerning the role of dipropyleneglycol as substituent of 1,4-butanediol on the physical properties of thermoplastic polyurethanes .In this order, authors reports analysis by Fourier-transform infrared spectroscopy (FTIR), UV-VIS spectroscopy, small-angle X-ray scattering (SAXS), differential scanning calorimetry (DSC) and mechanical properties. The discussions are well structured. In the revised manuscript, the following comments are necessary to be included:”
- i) what is the physical rationale for using 3 components in the deconvolution of Figures S1a-d and 4 components in the case of Figure S1e?
Answer to Reviewer: Thank you for your contribution. The use of 3 components in deconvolution is due to the fact that the contributions of the free carbonyl group, dipole-bonded and hydrogen bonded are taken into account. This is already well described in the literature:
-Niemczyk, A.; Piegat, A.; Olalla, Á.S.; El Fray, M. New approach to evaluate microphase separation in segmented polyurethanes containing carbonate macrodiol. European Polymer Journal 2017, 93, 182-191.
-Favero, D.; Marcon, V.; Figueroa, C.A.; Gómez, C.M.; Cros, A.; Garro, N.; Sanchis, M.J.; Carsí, M.; Bianchi, O. Effect of chain extender on the morphology, thermal, viscoelastic, and die-lectric behavior of soybean polyurethane. Journal of Applied Polymer Science 2021, 138, 50709.
- ii) Similar question for Figure S2
iii) Authors should to show the FTIR spectra in the 400-4000 cm-1 spectral range and then to focus the discussion on the spectral range 1600-1800 cm-1. A comment concerning other changes observed in the spectral range 400-4000 cm-1 will be welcome.
Answer to Reviewer: Thank you for your contribution. The FTIR data was included in the supplementary files and commented on in the article
Reviewer 3 Report
Comments to the Authors
This manuscript demonstrates that the authors have synthesized TPUs by reactive extrusion where the role of DPG and BDO were demonstrated in regards to their morphological properties and transparency. Overall the manuscript presented well, and the technique and characterization of the results achieved indicate that the method is quite suitable and in fact could be transformative in the morphology, thermal, and physicochemical properties of the synthesized TPUs. Finally, a few minor reviewer comments are provided.
i) In line 21, 4,4'-diphenylmethane (MDI), should be 4,4′-diphenylmethane diisocyanate (MDI).
ii) The name of the synthesized TPUs samples should be systematic. I would like such is “BDO/DPG-XX/XX” rather than XX/XX-BDO/DPG. The authors used both kinds of names which created confusion.
iii) The authors used the term BDO/DPG but in the conclusion, it is BD/DPG.
iv) I would like if a table can be added that explains the formulation, composition etc. of the synthesized TPUs.
v) A schematic diagram will be helpful if it explains the whole study in a single image or scheme.
vi) In Figure 4b, please change the color of the light green curve to dark green for a better visual appearance.
Author Response
Dear Reviewer,
On behalf of the authors, we thank you for your contributions.
“This manuscript demonstrates that the authors have synthesized TPUs by reactive extrusion where the role of DPG and BDO were demonstrated in regards to their morphological properties and transparency. Overall the manuscript presented well, and the technique and characterization of the results achieved indicate that the method is quite suitable and in fact could be transformative in the morphology, thermal, and physicochemical properties of the synthesized TPUs. Finally, a few minor reviewer comments are provided.”
- In line 21, 4,4'-diphenylmethane (MDI), should be 4,4′-diphenylmethane diisocyanate (MDI).
Answer to Reviewer: Thank you for your contribution. The manuscript was improved.
- The name of the synthesized TPUs samples should be systematic. I would like such is “BDO/DPG-XX/XX” rather than XX/XX-BDO/DPG. The authors used both kinds of names which created confusion.
Answer to Reviewer: Thank you for your contribution. The nomenclature was modified according to reviewer suggestion.
- The authors used the term BDO/DPG but in the conclusion, it is BD/DPG.
Answer to Reviewer: Thank you for your contribution. The manuscript was improved.
- I would like if a table can be added that explains the formulation, composition etc. of the synthesized TPUs.
Answer to Reviewer: Thank you for your contribution. The information is described in the experimental session
- A schematic diagram will be helpful if it explains the whole study in a single image or scheme.
Answer to Reviewer: Thank you for your contribution. Because the journal only gives 5 days to make the corrections, this inclusion is not feasible.
- vi)In Figure 4b, please change the color of the light green curve to dark green for a better visual appearance.
Answer to Reviewer: Thank you for your contribution. The color in figure was improved.
Round 2
Reviewer 2 Report
I recommend this paper to be accepted for publication in the present form in the Polymers journal.